# Towards a Diagnosis of Cardiac Amyloidosis: Single Center Experience with ^99m^ Technetium Pyrophosphate Planar Imaging and Opportunities for Standardization of Diagnostic Workflow [note 1]

**DOI:** 10.3390/medicina59020378

**Published:** 2023-02-16

**Authors:** Mariam Saleem, Besher Sadat, Meredith Van Harn, Karthikeyan Ananthasubramaniam

**Affiliations:** 1Department of Cardiovascular Medicine, Ascension Providence Southfield Hospital, Southfield, MI 48075, USA; 2Department of Cardiovascular Medicine, University of Texas Medical Branch of Galveston, Galveston, TX 77555, USA; 3Department of Public Health Sciences, Henry Ford Health System, Detroit, MI 48202, USA; 4Heart and Vascular Institute, Henry Ford West Bloomfield Hospital, West Bloomfield, MI 48322, USA

**Keywords:** cardiac amyloidosis, transthyretin cardiac amyloidosis, light chain amyloidosis, technetium pyrophosphate scan, restrictive cardiomyopathy

## Abstract

*Background and Objectives*: Cardiac amyloidosis is a disorder caused by amyloid fibril deposition in the extracellular space of the heart. Almost all forms of clinical cardiac amyloidosis are transthyretin amyloidosis (ATTR) or light chain amyloidosis. ^99m^ technetium pyrophosphate (^99m^Tc PYP scan) has changed the landscape of the non-biopsy diagnosis of ATTR cardiac amyloidosis (ATTR-CA) by providing remarkably high diagnostic accuracy. We examined our experience with PYP scans in patients undergoing workup for ATTR-CA and evaluated the diagnostic workflow in patients with intermediate PYP scan results. *Materials and Methods*: Retrospective chart review study in which we analyzed data of 84 patients who underwent c-99m pyrophosphate (PYP) SPECT scan for the diagnosis of ATTR-CA from 2017 till 2021 at our institution. We identified three groups: Low uptake (PYPL uptake ratio < 1.2 + visual grade 1/0), *n* = 30, Intermediate uptake (PYPI uptake ratio 1.2–1.49, visual grade 2/3), *n* = 25 and High uptake (PYPH uptake ratio ≥ 1.5 + visual grade 2/3), *n* = 29. We reviewed patients’ demographics, medical histories, echo parameters and diagnostic testing including light chain analysis, cardiac magnetic resonance results, and biopsies. *Results*: Mean patients’ age was 73, male-to=female ratio 3:1, 59% of patients were African American. Cardiovascular comorbidities, cardiac biomarkers (BNP and Troponin) and amyloid-related neuropathy were similar in all groups. A statistically significant difference in septal thickness/posterior wall thickness and final diagnosis were found between the groups. The distribution of overall diagnostic testing ratios for the PYPI group included serum protein electrophoresis 92%, urine protein electrophoresis 65%, free light chain 80%, CMR 32%, tissue biopsy done in 20% and BM biopsy in 16%, which are similar to the ratios of other groups. Overall, 25% (*n* = 5, 4 TTR-CA and 1 AL Amyloid) of patients in the PYPI group had a final diagnosis of CA established with additional testing (*p* = 0.001 vs. other groups). *Conclusions*: The ^99m^PYP scan is an accurate noninvasive test for cardiac ATTR-CA. Importantly, 25% of the PYPI group had a final diagnosis of ATTR-CA reiterating that diagnosis needs to be pursued in PYPI cases based on clinical suspicion. Routine evaluation and exclusion of light chain disease and establishing a consistent workflow for amyloid diagnosis and continued education for technologists and readers of PYP scans is key to a successful amyloidosis workup.

## 1. Introduction

Amyloidosis is an infiltrative disease characterized by deposition of amyloid fibrils within the extracellular tissue of one or multiple organs. It is a systemic disease that can involve any organs in the body. Amyloidosis usually presents with a wide range of clinical signs and symptoms including but not limited to poor appetite, early satiety, abdominal distension, hepatomegaly, nephrotic syndrome, peripheral neuropathy, carpel tunnel syndrome and others [1]. Involvement of the heart, cardiac amyloidosis (CA), is recognized as one of the most common causes of restrictive cardiomyopathy and heart failure [2,3,4]. Almost all forms of clinical cardiac amyloidosis are transthyretin amyloidosis (ATTR-CA; wild type and hereditary) or light chain cardiac amyloidosis [2]. ATTR-CA occurs when the native tetrameric form of TTR protein, pathologically separates to form amyloid fibrils that deposit in different organs while AL-CA occurs when an abnormal clone of plasma cells produces a light chain that is susceptible to misfolding and forming amyloid fibrils [5].

Patients with ATTR-CA usually experience a slow and progressive disease process leading to debilitating symptoms related to heart failure like exercise intolerance and fatigue, which results in decreased functional capacity, poor quality of life, and eventual death [6]. Cardiac amyloidosis is an underdiagnosed cause of cardiomyopathy. Despite the increased awareness of this condition, the diagnosis is usually delayed. This is due to multiple factors including nonspecific presenting symptoms that can be attributed to other causes and uneven access to noninvasive imaging modalities that are required for the diagnosis. Early detection is imperative as it impacts the management and improves disease outcome given treatments are more effective in early stages of the disease [3,4,7]. Recent advances in cardiac imaging have led to early diagnosis of ATTR-CA with the use of echocardiography and specifically strain analysis patterns (relative apical sparing), cardiac magnetic resonance imaging (MRI) with use of gadolinium and parametric imaging and finally ^Tc99m^ pyrophosphate scan (PYP), which has very high sensitivity and specificity; we can establish the diagnosis of ATTR-CA in the absence of tissue biopsy [8,9,10]. The ^Tc99m^PYP scan has been a game-changing imaging modality and importantly widely available nuclear imaging test for the diagnosis of ATTR-CA. It has remarkably high diagnostic accuracy with a sensitivity of 92.2% and specificity of 95.4%, and obviates the need for cardiac biopsy in most cases [9,10,11]. There has been a significant increase in the utilization of the ^Tc99m^PYP scan recently given its high sensitivity and specificity for ATTR-CA, and it has been incorporated as a key test in the workup. An important step in interpreting the scan results lies in conclusive evaluation and exclusion of light chain disease with monoclonal protein screening and immunofixation to rule out AL-CA and applying SPECT/CT imaging to establish definitive myocardial isotope uptake, particularly in inconclusive ^Tc99m^PYP scans [12,13,14,15]. Previous studies had shown that the quantitative and semiquantitative uptake intensity of ^Tc99m^PYP is associated with all-cause mortality, as well as all-cause mortality or heart failure hospitalization [16], and the use of high sensitivity troponin measurement along with ^Tc99m^PYP significantly increases the diagnostic rate of ATTR-CA [17].

The aim of this study was to study our early experience with planar ^Tc99m^PYP imaging and how clinicians utilized results of the planar imaging towards the diagnosis of CA. Hence, we reviewed ^Tc99m^PYP scans performed for patients undergoing workup for CA at our institution and examined the diagnostic workflow in patients with suspected CA utilizing this imaging modality focusing on the intermediate planar ^Tc99m^PYP scan results.

## 2. Materials and Methods

### 2.1. Study Population

We conducted a retrospective chart review study that enrolled adult patients (18 years and older) who underwent ^Tc99m^ pyrophosphate (PYP) scan for the diagnosis of CA from 2017 until 2021 at our institution.

### 2.2. Data Collection

The Henry Ford Health Institutional Review Board reviewed and approved the study protocol (#IRB13570). Eighty-four patients had ^Tc99m^PYP scan done in the selected time period. We reviewed and collected patients’ demographics including age, sex and race; medical histories including history of heart failure, atrial fibrillation, cerebrovascular accidents, carpal tunnel syndrome, spinal stenosis and peripheral neuropathy; admissions for heart failure exacerbation and arrhythmia, biomarkers (Troponin and B-Type natriuretic peptide); electrocardiogram and echocardiography results including ejection fraction, diastolic dysfunction grade, diastolic septal (IVSd) and posterior wall thickness (PWd), diagnostic testing including serum protein electrophoresis, urine protein electrophoresis, free light chains, cardiac magnetic resonance imaging, the ^Tc99m^ pyrophosphate (PYP) scan planar uptake ratio and tissue biopsies. Standard cine MRI images, delayed gadolinium enhancements and look-locker sequence were used in the imaging analysis. Biopsy specimens were analyzed by the Department of Pathology using Congo red staining and mass spectrometry.

#### ^Tc99m^PYP Planar Scintigraphy

Planar cardiac imaging was performed with a dual-headed camera. Fifteen millicuries of ^Tc99m^PYP isotope was injected and a 1 h incubation period was used for planar imaging. Anterior and lateral planar images were obtained for approximately 750,000 counts and images were interpreted using both a visual score (0 = no cardiac uptake, 1 = cardiac uptake less than bone, 2 = cardiac uptake equal to bone and 3 = cardiac uptake greater than bone) as shown in Figure 1. A quantitative analysis was also done using in the identical circular region of interest over the heart and contralateral lung to calculate the heart/contralateral (H/CL) ratio.

Patients were divided into three groups based on their visual grading and H/CL ratio [18,19]; Low uptake (PYPL with uptake ratio < 1.2 + visual grade 1/0), *n* = 30, Intermediate uptake (PYPI with uptake ratio 1.2–1.49+ visual grade 2/3), *n* = 25 and high uptake (PYPH uptake ratio > 1.5 + visual grade 2/3), *n* = 29. This study cohort primarily represents the use of planar ^Tc99m^PYP imaging for interpretation and reporting. Technologists proceeded with SPECT only if the planar H/CL ratio was in the intermediate range (1.2–1.49) and felt to be inconclusive per the reader. Hence, the use of SPECT in this study was very selective. For the purposes of this study, a blinded and independent re-review of all planar ^Tc99m^PYP scans was done and scored by a level III nuclear reader (KA).

### 2.3. Statistical Analysis

All continuous data are presented using means, medians, minimums, 25th percentiles, 75th percentile maximums, and standard deviations, while categorical variables are presented using counts and column percentages. All variables are compared between PYP groups using Kruskal-Wallis tests due to small sample size and violation of normality assumptions, while categorical variables are compared between groups using chi-square or Fisher’s exact tests based on expected cell count. Statistical significance was set at *p* < 0.05. Independent re-review of all PYP scans was done by a level III nuclear reader (KA).

All analyses are performed using SAS 9.4 (SAS Institute Inc., Cary, NC, USA).

## 3. Results

Data on 84 patients who underwent a ^Tc99m^PYP scan were analyzed. The overall mean age of the study group was 73 (mean age 67 in PYPL group, 71 PYPI and 79 in PYPH, *p* = 0.003); male-to-female ratio 3:1; 59% of patients were African American, 35% Caucasian and 6% of other racial backgrounds. Eighty three percent of patients had the diagnosis of heart failure by history at the time of the ^Tc99m^PYP scan, 42% had the diagnosis of atrial fibrillation, 17% had stroke, 20% had a history of carpel tunnel syndrome, 12% bilateral carpal tunnel syndrome, 12% lumber stenosis and 20% peripheral neuropathy. Forty percent of patients had a history of admission due to decompensated heart failure and 14% of patients had an admission for arrhythmia within 3 months of performing a ^Tc99m^PYP scan. Twenty-nine percent of patients met the criteria for low-voltage EKG. A summary of the analyzed patients’ characteristics is listed in (Table 1).

There was no statistically significant differences between the three groups in terms of cardiovascular comorbidities, cardiac biomarkers (BNP and troponin), amyloid-related neuropathy or previous admission history. A statistically significant difference in IVSd and PWd thickness were found between the groups (mean IVSd 1.42 cm in PYPL group, 1.31 cm in PYPI uptake and 1.72 cm in PYPH group, *p* ≤ 0.001, mean PWd of 1.32 cm in the PYPL group, 1.23 cm in the PYPI uptake and 1.59 cm in the PYPH group, *p* ≤ 0.001). The ejection fraction was noted to be significantly different between the three groups with a mean EF of 54% in the PYPL, 55% in the PYPI and 46% in the PYPH group, *p* = 0.04.

Diagnostic labs including serum and urine protein electrophoresis, serum and urine free light chains, cardiac MRI and tissue biopsies were performed in similar rates between the three groups (Table 2). AL amyloid was ruled out using monoclonal protein assay with free light chain determination, and serum and urine immunofixation and electrophoresis (SPEP and UPEP). SPEP was in 84% of patients and UPEP in 66.7%, and urine free light chain in 73%. However, no statistically significant difference was found between the groups. The overall use of cardiac MRI was 33% (26 patients); 19 of them were positive for CA.

Genetic testing for the TTR mutation was done in 50% of patients with confirmed TTR mutation (13 out of 26 patients). The majority of patients (95%) had a val112Ile mutation. There was one patient in the low uptake group that was found to have AL amyloid through bone marrow biopsy.

Of note, the PYPI group had the lower values of IVSd and PWd thickness compared to the other groups. To further clarify if these patients had CA, an array of diagnostic testing was done in this PYPI group; SPEP 92%, UPEP 65%, free light chains 80%, MRI 32%, tissue biopsy in 20% and bone marrow biopsy in 16%, which was comparable to rates of diagnostic testing in the other 2 groups. With the additional information, 25% of patients in the PYPI group reached a final diagnosis of amyloid (4 ATTR and 1 AL), which was statistically significant in comparison with the other groups, *p* ≤ 0.001 (Table 2). Our independent re-review of all ^Tc99m^PYP scans also resulted in identification of 2 cases, which was reclassified to the PYPL group.

## 4. Discussion

This work analyzed our early experience of planar ^Tc99m^PYP imaging at a single tertiary care center examining the diagnostic workflow of a patient with suspected CA, and categorizing patients according to the planar uptake ratios of the ^Tc99m^PYP scan into three groups.

Cardiac uptake of ^99mTc^PYP evaluation using a semi-quantitative visual scoring method in relation to bone uptake has been recommended given its high sensitivity and specificity to identify ATTR cardiac amyloidosis and distinguish it from AL amyloid. Recent ASNC/AHA/ASE/EANM/HFSA/ISA/SCMR/SNMMI Expert Consensus Recommendations for Multimodality Imaging in Cardiac Amyloidosis published in 2021 has recommended to include visual assessment of cardiac uptake in PYP scan interpretation^.^ Other interpretation methods include semi-quantitative evaluation by the heart/contralateral lung uptake [H/CL] ratio at 1 h and 3 h, and myocardial accumulation by single-photon emission computed tomography [SPECT]) [20,21]. A previous multicenter study reported that 1-h planar PYP images showed high sensitivity, while 3-h planar PYP images showed high specificity for the diagnosis of ATTR-CA [22,23]. Since the inception of planar ^Tc99m^PYP, imaging experience has accumulated and the incremental value of SPECT to clarify inconclusive cases and distinguish myocardial uptake from the blood pool has been realized. Hence, SPECT imaging can help reduce the rate of misdiagnosis and should be regularly performed [23].

The main findings of this study can be summarized as follows: First, ^Tc99m^PYP scan is a key diagnostic tool with high sensitivity and specificity for the diagnosis of ATT-CA when combined with conclusive negative light chain analysis for AL-CA. Our study showed that 93% of patients in the ^Tc99m^PYPH group were ultimately diagnosed with ATTR-CA, which is consistent with sensitivity reported in previous studies [9,10,11,14,17]. Second, increased IVSd and PWd thickness are key and readily available echocardiogram findings that should alert the physician on pursuing further testing for CA. Third, although the presence of positive biomarkers may trigger a suspicion of CA, the levels of cardiac biomarkers should not influence the decision on pursuing further cardiac imaging for CA; and finally, 25% of patients with ^Tc99m^PYPI scan uptake were reclassified with further diagnostic testing reiterating the importance of ensuring comprehensive work in inconclusive cases when CA suspicion is present.

There was no statistically significant difference between the three groups in terms of cardiovascular comorbidities or cardiac biomarkers, and hence medical history and clinical characteristics alone, and biomarkers appear insufficient to predict who will have a PYPH scan. We also found that diagnostic work up including serum and urine protein electrophoresis, urine free light chains, cardiac MRI, and biopsies were performed at similar rates between the three groups but were overall not consistently performed in all groups. This suggests the need for standardization of workup for CA and more systematic education of clinical providers as to how to approach suspected patients with CA (Table 2).

Statistically significant differences in IVSd and PWd thickness were noted between the groups with the PYPH group showing the highest values. Of note, a recent study evaluating the role of echo parameters in predicting PYP scan positivity showed similar findings. In this study inferolateral (posterior) wall thickness >14 mm along with basal longitudinal strain were the best predictors of PYP positivity [24].

A group of particular interest is those with PYPI (PYP uptake ratio 1.2–1.49+ visual grade 2/3 uptake on planar), in which diagnosis might be missed if workup is not complete. We found that patients in the PYPI group had the lower values of IVSd and PWd thickness compared to the other groups. Further diagnostic testing to evaluate for CA in this group was variable as noted above. Twenty five percent of patients in the intermediate group had a final diagnosis of CA after further investigation, which was statistically significant compared with the other groups, which reiterates the specific need to pursue CA workup in indeterminate ^Tc99m^PYP scans.

## 5. Limitations

The major limitation of the study pertains to the retrospective single-center setup and small sample size. The study populations were predominantly African American, limiting its generalizability. Also, this study was done in the early phase of our program when SPECT was only selectively done so SPECT data was not routinely available. The PYPI group particularly would benefit from SPECT imaging to help confirm or exclude myocardial uptake. As planar imaging alone is insufficient in a proportion of cases, we have hence modified our protocol and currently routinely perform SPECTCT imaging along with planar ^Tc99m^PYP imaging and report planar and SPECT data. Currently, our protocol routinely performs planar and SPECT CT, which enhances diagnostic capabilities in intermediate cases and avoids false positives related to blood pool uptake causing elevations in planar ratios.

## 6. Conclusions

The ^Tc99m^PYP scan is an accurate noninvasive test for cardiac ATTR -CA. Importantly, 25% of the PYPI group had a final diagnosis of ATTR -CA, reiterating that early TTR-CA or AL need to be excluded by conclusive further evaluation. Our study predated SPECT-CT imaging and identifies a better need for standardization for light chain evaluation and a consistent workflow for CA diagnosis. Since the completion of this study, routine SPECT CT is now part of our protocol, and a standardized workflow and diagnostic algorithm has been implemented for CA diagnosis.

## Figures and Tables

**Figure 1 medicina-59-00378-f001:**
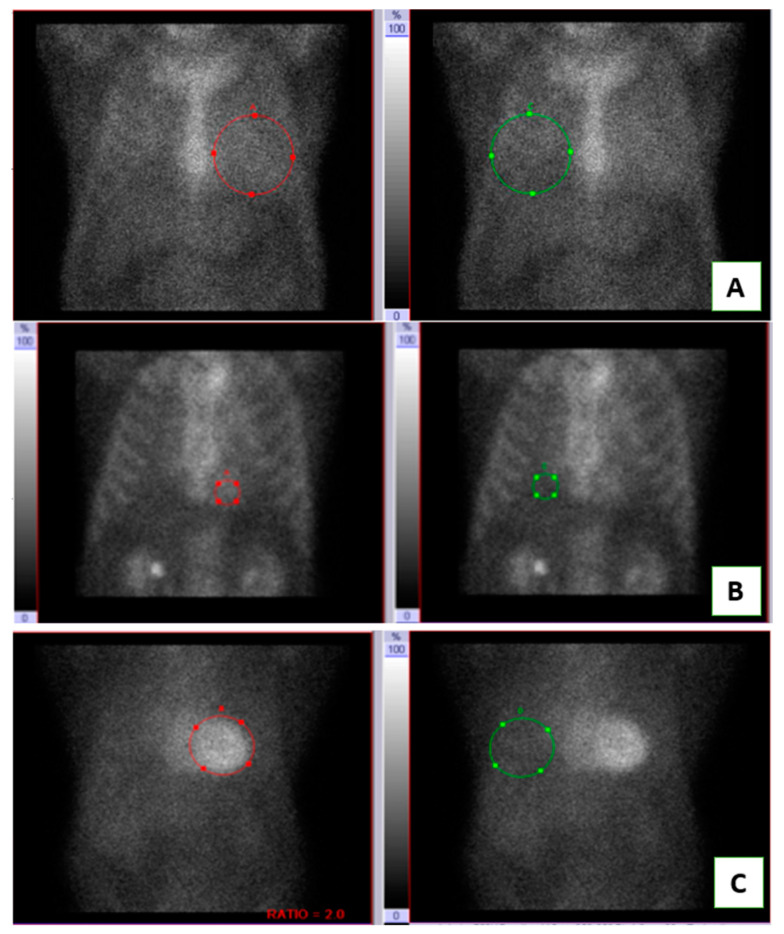
An example of planar PYP scan images of low, intermediate, and high uptake groups: (**A**) Low uptake with Heart/Contralateral lung (H/CL) ratio 1.1, (**B**) Intermediate uptake with H/CL ratio = 1.3, (**C**) High uptake with H/CL ratio = 2.0. Red: heart uptake, Green: contralateral lung uptake.

**Table 1 medicina-59-00378-t001:** Descriptive statistics.

Variable	Level	*n* (%) = 84
Age	Mean	73.01
Sex	Female	20 (23.8)
	Male	64 (76.2)
Race	AA	49 (59.0)
	White	29 (34.9)
	Other	5 (6.0)
History of heart failure		69 (83.1)
History of atrial fibrillation		34 (41.5)
History of cardiovascular accidents		14 (17.1)
History of carpal tunnel syndrome		16 (19.5)
Bilateral carpal tunnel syndrome		9 (12.0)
Lumbar stenosis		10 (12.2)
Neuropathy		17 (20.7)
Admission for congestive heart failure exacerbation		32 (40.5)
Admission for arrhythmia		11 (13.9)
EKG low voltage		23 (29.5)
Serum protein electrophoresis		67 (84.8)
Urine protein electrophoresis		52 (66.7)
Urine light chains test		58 (73.4)
Cardiac MRI done		26 (32.9)
Tissue biopsy		16 (20.5)
Biopsy type	Fat pad	9 (60.0)
	Endomyocardium	6 (40.0)
Bone marrow biopsy		10 (12.7)
Genetic testing done		22 (27.8)
Final diagnosis of cardiac amyloid		33 (40.7)
PYP group	Low	30 (35.7)
	Intermediate	25 (29.8)
	High	29 (34.5)

**Table 2 medicina-59-00378-t002:** Univariate comparisons between PYP groups.

			PYP Group			
Covariate	Statistics *n* (Col %)	Level	Low *n* = 30	Intermediate *n* = 25	High *n* = 29	*p*-Value
History of heart failure	*n* (Col %)	Yes	24 (80)	20 (80)	25 (89.29)	0.615
History of Afib			8 (27.59)	12 (48)	14 (50)	0.167
History of CVA			5 (17.24)	6 (24)	3 (10.71)	0.471
History of CTS			3 (10.34)	5 (20)	8 (28.57)	0.221
Bilateral CTS			1 (3.85)	3 (13.64)	5 (18.52)	0.266
Lumbar stenosis			3 (10.34)	4 (16)	3 (10.71)	0.836
Neuropathy			7 (24.14)	4 (16)	6 (21.43)	0.758
Admission for CHF			10 (38.46)	9 (36)	13 (46.43)	0.718
Admission for arrhythmia			4 (15.38)	3 (12)	4 (14.29)	1.000
BNP within 3 months of PYP scan	Mean		1178.84	1021.09	802.69	0.419
Troponin within 3 months of PYP scan	Mean		89.64	84.42	141.07	0.421
EF % on echo	Mean		0.54	0.55	0.46	0.040
Septal IVSD thickness on echo	Mean		1.42	1.31	1.72	<0.001
Posterior wall thickness on echo	Mean		1.32	1.23	1.59	<0.001
Diastolic dysfunction grade	*n* (Col %)	None	1 (4.76)	2 (9.52)	0 (0)	0.210
	*n* (Col %)	Mild	7 (33.33)	7 (33.33)	4 (17.39)	
	*n* (Col %)	Moderate	6 (28.57)	7 (33.33)	5 (21.74)	
	*n* (Col %)	Severe	7 (33.33)	5 (23.81)	14 (60.87)	
EKG low voltage			7 (26.92)	6 (25)	10 (35.71)	0.658
Serum protein electrophoresis done			20 (76.92)	23 (92)	24 (85.71)	0.334
Urine electrophoresis done			16 (64)	16 (64)	20 (71.43)	0.800
Urine light chains test done			16 (61.54)	20 (80)	22 (78.57)	0.245
Cardiac MRI done						0.104
			5 (19.23)	8 (32)	13 (46.43)	
Tissue biopsy done			4 (16)	5 (20)	7 (25)	0.718
Biopsy type		Fat pad	1 (20)	3 (75)	5 (83.33)	0.111
		Myocardial	4 (80)	1 (25)	1 (16.67)	
Final diagnosis of CA		Yes	1 (3.7)	5 (20)	27 (93.1)	<0.001

## Data Availability

Not applicable.

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
