# Peer review of "Towards a Diagnosis of Cardiac Amyloidosis: Single Center Experience with 99m Technetium Pyrophosphate Planar Imaging and Opportunities for Standardization of Diagnostic Workflow†"

_medicina, 2023, doi:10.3390/medicina59020378_

Round 1

Reviewer 1 Report

In the present study Dr Saleem and coworkers presented retrospectively data of patients undergoing planar TC 99m PYP SPECT for the detection of TTR amyloidosis. They compared the results of the exams performed, according to the grading of cardiac PYP uptake

Comments:

1. Please define the abbreviations SPEP and UPEP presented in the abstract

2. Why did the authors use the Peruggini score for the grading of PYP uptake? Please provide references of other studies using the same method of PYP cardiac uptake grading

3. Could intermediate cases of PYP uptake be better defined if SPECT immediately after planar images was performed?

4. How the authors can explain that there was no difference in the rate of MRI or biopsy performed between the intermediate and high uptake cases? According to current algorithms, high uptake and normal chain electophoresis has high sensitivity for the diagnosis of ATTR

5. Please provide an image depicting cases of low, intermediate and high uptake cases

6. According to the authors opinion, how this work can contribute to the present algorithms for the diagnosis of ATTR?

Reviewer 2 Report

Although I think it is very valuable to report field experience with new diagnostic test, I have a few concerns.

- why did not all patients have proper rule out of AL amyloydosis with serum and urine tests? 

- how was AL amyloidosis ruled out in these patients?

- was genetic testing done in patients with confirmed ATTR-amyloidosis, what were the results? Especially of interest in the low uptake group, where there was a patient with CA (but what type?).

- what analysis was used for evaluation of the biopsies? mass spectometry?

- in almost 33% of patients an MRI was done, what were the results? what sequences where done, ECV calculation, T1 / T2 mapping?

- concerning the PYP scintigraphy? what was the injected activity? what was the time between injection and scan? Why was the re-read only done on the planar imagnig? How was misinterpretation of bloodpool activity (what is one of the main reseason for doing a SPECT(CT)) avoided?

Round 2

Reviewer 1 Report

My comments have been adequately addressed

Author Response

All the comments have been addressed

Thank you for your time in reviewing our paper.

Reviewer 2 Report

I still have a major concern about the diagnostic work-up is not complete in all patients. It is not clear what alternative methods where used is ie SEP or UEP was missing. Patients with incomplete diagnostic work-up should be excluded, because a positive PYP scan does not rule out AL amyloidosis!

I think it is also essential to perform at least SPECT when using PYP to avoid misinterpretation of bloodpool activity and if that is missing, those patients should be excluded from the analysis.

Author Response

We thank the reviewer for this important comment and agree that the current standard of amyloid PYP imaging should include SPECT CT and complementary planar imaging. However, as mentioned in the introduction and limitations the goal of our study here was to look back at our early experience when we launched PYP planar imaging where SPECT was used only in selective cases based on reader decision and to identify the process of workup and areas in need for more standardized workflow.
Patients underwent light chain analysis and serum and urine immunofixation electrophoresis to exclude AL and when the PYP scan was intermediate in this setting as illustrated underwent a variety of additional testing. We have since instituted a standardized order set in our system for light chain disease evaluation (monoclonal protein screen pathway) along with performing routine PYP SPECT-CT imaging based on findings shown in this retrospective study